# Crocodilepox Virus Evolutionary Genomics Supports Observed Poxvirus Infection Dynamics on Saltwater Crocodile (*Crocodylus porosus*)

**DOI:** 10.3390/v11121116

**Published:** 2019-12-02

**Authors:** Subir Sarker, Sally R. Isberg, Jasmin L. Moran, Rachel De Araujo, Nikki Elliott, Lorna Melville, Travis Beddoe, Karla J. Helbig

**Affiliations:** 1Department of Physiology, Anatomy and Microbiology, School of Life Sciences, La Trobe University, Bundoora, VIC 3086, Australia; 2Centre for Crocodile Research, Noonamah, NT 0837, Australia; sally@crocresearch.com.au (S.R.I.); research@crocresearch.com.au (J.L.M.); 3School of Psychological and Clinical Sciences, Charles Darwin University, Darwin, NT 0909, Australia; 4Berrimah Veterinary Laboratory, Northern Territory Government, Darwin, 0801 Northern Territory, Australia; Rachel.DeAraujo@nt.gov.au (R.D.A.); Nikki.Elliott@nt.gov.au (N.E.); Lorna.Melville@nt.gov.au (L.M.); 5Department of Agriculture Sciences, School of Life Sciences, La Trobe University, Bundoora, VIC 3086, Australia; T.Beddoe@latrobe.edu.au

**Keywords:** saltwater crocodilepox virus, infection dynamics, complete genome, evolution, genetic recombination

## Abstract

Saltwater crocodilepox virus (SwCRV), belonging to the genus *Crocodylidpoxvirus*, are large DNA viruses posing an economic risk to Australian saltwater crocodile (*Crocodylus porosus*) farms by extending production times. Although poxvirus-like particles and sequences have been confirmed, their infection dynamics, inter-farm genetic variability and evolutionary relationships remain largely unknown. In this study, a poxvirus infection dynamics study was conducted on two *C. porosus* farms. One farm (Farm 2) showed twice the infection rate, and more concerningly, an increase in the number of early- to late-stage poxvirus lesions as crocodiles approached harvest size, reflecting the extended production periods observed on this farm. To determine if there was a genetic basis for this difference, 14 complete SwCRV genomes were isolated from lesions sourced from five Australian farms. They encompassed all the conserved genes when compared to the two previously reported SwCRV genomes and fell within three major clades. Farm 2′s SwCRV sequences were distributed across all three clades, highlighting the likely mode of inter-farm transmission. Twenty-four recombination events were detected, with one recombination event resulting in consistent fragmentation of the P4c gene in the majority of the Farm 2 SwCRV isolates. Further investigation into the evolution of poxvirus infection in farmed crocodiles may offer valuable insights in evolution of this viral family and afford the opportunity to obtain crucial information into natural viral selection processes in an in vivo setting.

## 1. Introduction

The *Poxviridae* family are large double-stranded DNA viruses with a complex structure and a broad linear genome ranging from 128 to 365 kbp. The evolutionary origin of poxviruses is still ill-defined, however, it is believed that their genomes have evolved over thousands of years through both gene gain and loss, mainly through horizontal gene transfer and gene duplication events [1,2]. The saltwater crocodilepox virus (SwCRV) belongs to the genus *Crocodylidpoxvirus*, a member of the subfamily *Chordopoxvirinae* in the family *Poxviridae* and is a known causative agent of poxviral lesions on Australian saltwater crocodile skin (*Crocodylus porosus*) [3,4]. Although the International Committee on Taxonomy of Viruses formally recognised crocodile poxviruses under the genus *Crocodylidpoxvirus* [5], at this stage, no taxonomic classification has been granted for SwCRV. A recent study suggested that there were likely two major SwCRV subtypes naturally circulating in the saltwater crocodile population [3] on one farm. However, relatively little is known about the origins, infection dynamics, genetic diversity and inter-farm genetic variability among the circulated SwCRV in saltwater crocodiles. In agreement with other crocodilian poxviruses [6,7], saltwater crocodilepox viruses are morphologically similar to orthopoxvirus virions, demonstrating a brick-like shape with rounded corners and a dumbbell-shaped central core and lateral bodies [3,4]. Furthermore, intracellular mature virions of SwCRV display the regular crisscross surface structure pattern, which is characteristic of parapoxvirus virions [3,4,7,8].

Poxvirus lesions have been reported in a number of different crocodilians, including *Caiman crocodilus fuscus, Caiman crocodilus yacare, C. porosus, Crocodylus johnstoni* and *Crocodylus niloticus* [6,7,9,10,11]. In *C. porosus*, poxvirus infection has been reported as a significant skin pathogen because if an individual is harvested with one or more poxvirus lesions, the lesion will result in an obvious defect on the finished leather product (4, 10). Moore et al. (4) described four stages of the poxvirus lesion development on *C. porosus* belly skins. The “early active” stage is characterised by lesions that are, on average 0.85 ± 0.29 mm^2^ grey-white foci with normal to pin-point keratin damage. As the lesion progresses into the “active” stage, there is an obvious enlargement of the lesion as the central plug (keratinocytes containing virus inclusion bodies) increases, compacting the underlying dermis and dislocating the overlying keratin. This enlargement continues until the central plug is expelled (“expulsion” stage) into the environment and the “healing” stage begins. Throughout these stages, histology reveals that poxvirus lesions do not breach the basement membrane layer of saltwater crocodile epidermis and given enough time they will heal without detriment to the quality of the finished leather product (4, 10). However, waiting for lesions to heal extends the production time of crocodiles, and, therefore, production costs, notwithstanding the risk of more lesions developing in the meantime (10). As such, poxvirus poses a substantial financial risk to Australian crocodile producers [4,12].

Even though SwCRV is an important pathogen of *C. porosus*, data regarding its evolutionary history, genetic diversity and molecular epidemiology are not sufficient due to the limited collection of only two complete genomes of SwCRV thus far. Therefore, this study was designed to firstly understand poxvirus infection dynamics, followed by developing a comprehensive sequence profile of a set of representative SwCRV genomes to identify the likely evolutional history, genetic diversity and inter-farm genetic recombination patterns across five different crocodile farms located in Northern Australia. In this study, 14 complete SwCRV genomes (12 SwCRV1 and 2 SwCRV2) were sequenced, assembled and annotated. Combined with the previous two SwCRV (3), these genomes represent a robust tool for studying the evolutionary history and genetic diversity of SwCRV and for identifying likely recombination events within SwCRV. This dataset may also offer valuable insights into the evolution of poxviruses as it represents sequence analysis of a group of highly related poxviruses in a unique environment where infection is constant and reoccurring.

## 2. Materials and Methods

### 2.1. Animal Sampling to Study Poxvirus Infection Dynamics

This study was conducted on 2 crocodile farms in the Northern Territory of Australia, and all animal sampling was conducted to comply with approved guidelines set by the Australian Code of Practice for the Care and Use of Animals for Scientific Purposes (1997) and approved by the Charles Darwin University Animal Ethics Committee (A16005) (17, February, 2016).

Crocodiles were sampled during both the dry and wet season for 2 consecutive years, as per Table 1. Hatchlings, grower and finishing pens [4] were randomly sampled to understand the occurrence of these lesions during the different production stages. At each sampling, 5 hatchling (7% of the total number of crocodiles in each pen), 10 grower (9%) and 10 finishing pen (10%) crocodiles were randomly selected from multiple randomly selected pens within each farm’s production stage to get an understanding of pox lesion prevalence across each farm (Table 1). The management of crocodiles was similar on both farms. Hatchlings were placed into pens for their first year, then they were moved into grower pens. There was some mixing of crocodiles at this time to minimize any size variation. Crocodiles remained in these pens until being moved into the finishing pens. Selection for finishing pens was dependent on size (135 cm+), as well as the number and severity of skin defects [11], including poxvirus lesions.

Smaller crocodiles (<1 m) were hand caught, whereas larger crocodiles (>1 m) were caught using electrical immobilisation [13]. The crocodile was then rolled onto its back and examined for the number and stages of characteristic poxvirus-lesions. A photograph of the full belly skin was taken as well as individual photos of lesions that were extruded for poxvirus PCR amplification (Figure 1). Prior to extrusion, the lesion was wiped with 70% ethanol and then extracted using a plastic pipette tip and stored in a sterile polystyrene tube at −20 °C until processing.

### 2.2. Extraction of DNA and PCR Screening for Poxvirus

Genomic DNA was isolated from the extruded lesion samples using a MagMAX-96 viral RNA isolation kit (Thermo Fisher Scientific, Waltham, MA, USA), according to the manufacturer’s instructions for animal tissue. As described previously by reference [4], PCR was performed using a HotStarTaq PCR Kit (Qiagen, Hilden, Germany), including 5 μL of DNA, 12.5 μL of 2X Master Mix, 1.25 μL (20 μM) of each forward and reverse primer and 5 μL of RNase-free water (poxvirus primers used were ORF99 (forward 5′-CATCCCCAAGGAGACCAACGAG, reverse 5′-TCCTCGTCGCCGTCGAAGTC). PCR was performed with an initial denaturation at 94 °C for 30 s, followed by 30 cycles at 94 °C for 30 s, 71 °C for 30 s and 72 °C for 60 s, with a final extension of 72 °C for 2 min. The PCR products were visualised on 2% agarose gels. All lesion samples collected were tested for poxvirus by PCR and random samples were sequenced for confirmation. For those lesions not found to be poxvirus, alternative causation was sought (for example *Dermatophilus* sp. [14] and the Kunjin strain of the West Nile virus (WNVKUN; [15]).

### 2.3. Statistical Analyses

The outcomes from the poxvirus PCR amplification provided confidence of lesions correctly classified as poxvirus. Thus, after the PCRs were completed, photographs of the crocodiles were re-examined, and lesions suspected not to be poxvirus were excluded from the dataset [14]. Data were analysed as binary data (presence/absence of poxvirus) as well as count data for the number of lesions in the different poxvirus stages on each belly skin [4] as follows.

Each crocodile was assigned either 0, if there was no evidence of poxvirus lesions on the belly skin, or 1 if characteristic poxvirus lesions were present. The binary trait was modelled using a GLMM (Generalised linear mixed model) in Genstat (version 17.1; VSN International Ltd., Oxford, UK) and explanatory factors included Farm (Farm 1 or Farm 2), Sampling (1, 2, 3 or 4), age categories (hatchling, grower or finishing pen) as well as body score (scale of 1 to 5 with 3 being ideal). All interactions were included, and a 5% significance level was chosen to evaluate the explanatory variable using backward elimination. Over-dispersion was allowed for in the model. All results were reported as back-transformed model-based means ± standard errors (SE).

For each crocodile, the number of lesions in each poxvirus stage was also recorded. The lesion stages were, as previously described [4], early active, active, expulsion and healing. As this was count data, a Poisson distribution was required in addition to using a GLMM in Genstat (version 17.1; VSN International Ltd., Oxford, UK) using the same explanatory factors as described for the binary analysis. In addition, to understand more about the infection dynamics, the lesions count’s in the other stages were also included as covariates in some models. Over-dispersion was allowed for in the model. As above, all interactions were included, and a 5% significance level was chosen. All results were reported as back-transformed model-based means ± standard errors (SE).

### 2.4. Virus Genome Sequencing and Analyses

A total of 10 exudative poxvirus lesions from the belly skin of juvenile saltwater crocodiles were sourced from Farms 1 and 2, including 2 lesions from Farm 1, whose SwCRV sequences were previously published (Genbank accession numbers MG450915 and MG450916). Additionally, a further 6 exudative poxvirus lesions from another 3 Northern Australian crocodile farms were included. DNA extraction from the collected samples was performed according to our previously published protocols [16,17]. Briefly, exudative pox lesions from individual crocodiles were aseptically dissected and mechanically homogenized in lysis buffer using disposable tissue grinder pestles and transferred into a 1.5 mL microcentrifuge tube (Eppendorf, Hamburg, Germany). The total genomic DNA (gDNA) was extracted using a ReliaPrep gDNA Tissue Miniprep System (Promega, Madison, WI, USA). Library preparation was conducted using one ng of total gDNA using the Illumina Nextera XT DNA Library Prep V3 Kit, according to our published protocol [3,18]. The quality and quantity of the prepared library were assessed using an Agilent Tape Station (Agilent Technologies, Santa Clara, CA, USA) by Genomic Platform, La Trobe University followed by paired-end sequencing on the Illumina MiSeq platform according to the manufacturer’s instructions. The sequence data were assembled according to our previously established protocols [3] using Geneious (version 10.2.2) and CLC Genomics workbench (version 9.5.4). The saltwater crocodilepox virus subtype 1 (SwCRV-1) was used as a reference genome for the annotation of all the SwCRV genomes sequenced in this study using our previously described methods [3].

### 2.5. Phylogenetic Analyses

Compete genome sequences under the genus *Crocodylidpoxvirus*, including 16 SwCRV (GenBank accession numbers-MG450915-16, MK903850- 63) and a Nile crocodilepox virus (GenBank accession number-DQ356948) [6], were aligned with MAFTT (Multiple *Alignment* using Fast Fourier Transform; version 7.388) [19] in Geneious. A selection of ~137 kbp core regions corresponding between CRV036 and CRV147 (large gaps removed) from the complete genome sequences of *crocodylidpoxvirus* were aligned with MAFTT (version 7.388) [19] in Geneious. Protein sequences of DNA polymerase genes (homologs to Molluscum contagiosum virus MC039L and Vaccinia virus E9L) were aligned with MAFTT (version 7.388) [19] in Geneious (version 10.2.2) under the BLOSUM62 scoring matrix. Pairwise similarities and distances were computed for the corresponding alignments using Geneious (version 10.2.2) and CLC Genomic Workbench (version 9.5.4). The maximum likelihood (ML) phylogenetic tree for complete genome sequences was obtained with PhyML [20] using a general-time-reversible model with gamma distribution rate variation and a proportion of invariable sites (GTR+G4) in Geneious. A ML tree for protein sequences of DNA polymerase gene was also constructed with PhyML [20] under the LG substitution model, and 1000 bootstrap resamplings were chosen to generate ML trees using tools available in Geneious (version 10.2.2).

Furthermore, analyses of the non-tree like evolutionary relationship amongst the SwCRV sequences were visualised using the neighbour-net algorithm using default parameters implemented in SplitsTree4 [21].

### 2.6. Recombination Analyses

Evidence of recombination amongst the SwCRV genome sequences were screened using the RDP [22], GENECONV [23], Bootscan [24], MaxChi [25], Chimaera [26], Siscan [27] and 3Seq [28] methods contained in the RDP4 program [29]. Recombination events that were detected by at least 3 of the methods described above with significant *p*-values (<0.05) were considered plausible recombinant events. Sequences that most closely resembled the parental sequences of recombinants were defined as either ‘minor parents’ or ‘major parents’ based on the size of the genome fragments that these sequences had contributed to the detected recombinants (with the major parent contributing the larger fragment and the minor parent the smaller).

To test that the detected recombination events did not arise from an assembly error due to the presence of different co-infecting SwCRV variants, the selected recombination events were further analysed using ML-based phylogenetics, NeighborNet trees and genetic distances. ML trees were constructed under the GTR substitution model, and 1000 non-parametric bootstrap resamplings were chosen in Geneious (version 10.2.2). Furthermore, analyses of non-tree-like evolutionary relationships amongst the selected recombination events were visualised with the NeighbourNet algorithm using default parameters implemented in SplitsTree4 [21]. To verify the statistical significance of the detected recombination events, a Phi test was conducted in SplitTree [30]. The recombination events were further visualised in more detail using Geneious software (version 10.2.2) to display variations/SNPs, and pairwise distances were computed for the corresponding alignments using the CLC Genomic Workbench (version 9.5.4).

## 3. Results

### 3.1. Prevalence and Infection Dynamics of Poxvirus

To assess the prevalence and pathogenesis of crocodile poxvirus lesions on two different farms with similar husbandry practices, crocodiles were assessed during both the dry and wet season for two consecutive years (Table 1). Using the defined poxvirus stages of Moore et al. [4], 82% of lesions were correctly identified as poxvirus lesions (Table 2). Fewer expulsion stage lesions could be sampled as the majority had already expelled their central plug, which contains the poxvirus DNA required for PCR detection, into the environment. For this very reason, no healing stage lesions could be confirmed by PCR. Expulsion lesions are very characteristic, thus 100% of the lesions sampled (*n* = 21) were confirmed as poxvirus. Active lesions also had high predictability of being confirmed for poxvirus (84%), with the mis-assigned lesions being confirmed as *Dermatophilus* sp. [14]. The early active lesions were the least successfully assigned, with only 67% confirming poxvirus. This poor assignment was biased by the first sampling period on Farm 1, whereby 33% of lesions collected were caused by WNVKUN, as first described by Isberg et al. [15]. Furthermore, 23.5% of early active lesions from both farms in sampling 1 were *Dermatophilus* sp. If sampling 1 was removed, correct assignment of early active lesions increased to 84.6%.

After eliminating lesions from the dataset that were not poxvirus based on PCR and re-examination of the belly skin photographs, the proportion of belly skins with poxvirus lesions are shown in Figure 2. Using a binary analysis, Farm 2 crocodiles were twice as likely to have poxvirus lesions compared to Farm 1 (2.02 ± 0.18; *p* < 0.001). Within both farms, hatchlings had the lowest probability of having poxvirus lesions on their belly skins. In comparison, grower crocodiles were seven times (7.02 ± 1.61; *p* < 0.001) more likely to have poxvirus lesions and finishing pens five times (5.08 ± 1.28; *p* < 0.001) more likely. There was no significant difference between sampling periods (*p* = 0.12; Figure 2) or crocodile body condition (*p* = 0.44) and there was no interaction between Farm and Age category (*p* = 0.12).

The count data revealed that Farm 2 crocodiles had significantly more early active (1.58 ± 0.21; *p* < 0.001; Figure 3A) and healing lesions (5.04 ± 0.64; *p* < 0.001; Figure 3D) than Farm 1. Sampling also had a significant effect on the observed number of each poxvirus lesion stage. Sampling 1 and 3 were just after Northern Australia’s dry season (cool nights and days with low humidity) and had the highest observed number of early active (sampling 1 only), active and expulsion lesions compared to those taken during the wet season (Sampling 2 and 4; very hot and humid; Figure 3).

In an attempt to better understand poxvirus infection dynamics, the number of lesions in the other poxvirus stages were also included as covariates in the analyses. In all cases, at least one other poxvirus lesion stage significantly affected the observed number of lesions at another stage in the infection cycle. There were more early active lesions observed on skins when active (1.27 ± 0.01), expulsion (1.11 ± 0.01) and healing (1.02 ± 0.004; Table 3) stage lesions were already present. The number of active lesions was higher when more early active lesions were developing (1.5 ± 0.03), as well as when more lesions were at the healing stage (1.02 ± 0.004). Only earlier stage lesions attributed to the number of expulsion lesions, while the number of healing stage lesions increased with more active stage lesions (1.13 ± 0.02) but not any of the other stages.

The crocodile total length was confounded within the age category. Thus, it was of interest to see if there was a bias not to move crocodiles with poxvirus lesions into finishing pens due to the higher costs of production. The relationship of poxvirus lesion count and total length was not linear, thus the data were categorized. Various models were evaluated and segregating total length into four 25 cm size categories was found to be the most appropriate when using the likelihood ratio tests (Figure 3). Interestingly, there appears to be a different infection strategy between the two crocodile farms. On Farm 1, early active (Figure 4A) and active (Figure 4B) lesions were highest in prevalence on the smaller crocodiles and decreased as the animal reached harvest size (*p* < 0.001). In contrast on Farm 2, the smaller animals had a lower prevalence, and the number of lesions increased as the crocodiles approach finishing size (*p* < 0.001). Expulsion lesions (Figure 4C) on Farm 1 also initially decreased, as per the early active and active lesions, but significantly increased to an average of 0.24 ± 0.07 lesions per belly skin in the 135+ cm size category, suggesting there was still a risk of increasing production time in these crocodiles. On Farm 2, there was a significant increase in the number of expulsion lesions in the 85–110 cm size group (*p* < 0.05), but these then stabilised at 0.34 ± 0.08 expulsion lesions in the larger size categories. The number of healing lesions (Figure 4D) on Farm 1 belly skins were not significantly different between size groups (*p* > 0.05) and were present at an average of 0.44 ± 0.21 healing lesions per skin on the 135+ cm size category. By vast contrast was the number of healing lesions (7.30 ± 1.41) present on Farm 2 animals as they approached finishing-size (135+ cm), which undoubtedly caused significant production delays, particularly given there were more early active (0.56 ± 0.14) and active poxvirus lesions (1.06 ± 0.25), which would also be required to go through the expulsion and healing stages before the animal could be harvested.

### 3.2. Characteristics of the SwCRV Genome Sequences

Our data indicate that poxvirus presentation and outcome may be varied amongst Australian farms despite similar farming practices. In order to assess the SwCRV genome distributions between farms, a total of 16, including 14 new complete SwCRV genomes and 2 previously published genomes [3], were interrogated (Table 4). The length of the SwCRV genomes sequenced in this study ranged from approximately 184 Kb to 187 Kb, with an average coverage range from 77.96× to 1905.90×. Similar to our previous study characterising SwCRV subtype-1 and -2 [3], we were able to group the 14 additional genomes into the two established subtypes by considering the nucleotide similarity percentage (Appendix A). The SwCRV subtype-2 comprised of a single isolate from Farm 1 (F1e; MG450916), Farm 2 (F2c; MK903855) and Farm 4 (F4a; MK903863) that demonstrated <98% nucleotide identity in comparison to SwCRV subtype-1 (Appendix A). In comparison, all other isolates comprising SwCRV subtype-1 were highly similar to each other when considering the nucleotide identities (>98%; Appendix A). The number of annotated genes identified in all 16 SwCRV genomes sequenced ranged from 211 to 218 (Table 4). There were no conserved genes missing and most of the absent genes encoded hypothetical proteins in comparison to SwCRV-1. Of particular interest was the significant variation in the gene encoding intracellular mature virus (IMV) A type inclusion-like protein P4c (SwCRV1-188), which showed multiple insertions/deletions and gene fragmentation (Figure 5). Importantly, except for one SwCRV isolate (F2c; MK903855), all other SWCRV isolates from Farm 2 showed the P4c gene to be fragmented (Figure 5).

### 3.3. Phylogenetic Cluster Definition and Sequence Similarities

ML phylogenetic analysis using ~137 kbp core region selected from complete viral genome sequences available in GenBank and isolated in the present study under the genus *Crocodylidpoxvirus* revealed that all the Australian SwCRV subtype-1 and -2 isolated from five different crocodile farms fell within three major clades (Figure 6A). In this phylogeny, clade-I comprised SwCRV-1 originating from the Farms 2, 3 and 5, whereas clade-II was mostly dominated by SwCRV-1 isolated from Farm 1 in addition to one isolate from Farm 2. Interestingly, an isolate that generated the first complete SwCRV-2 genome [3] phylogenetically grouped with two SwCRV-2 genomes isolated from Farms 2 and 4 (clade-III). The only other *crocodylidpoxvirus* genome sequence available (Nile crocodilepox virus) formed a separate clade (clade-IV) and did not show any close phylogenetic relationship with any SwcRV genome sequences isolated in this study, which was consistent with our previous observations [3]. Additionally, several farm dependent clades were observed within SwCRV genomes isolated from Farms 2, 3 and 5, to which no names have yet been assigned (Figure 6A). Furthermore, SwCRV genomes isolated from Farm 2 distributed across all the three clades, highlighting the likely mode of inter-farm SwCRV transmission. Whether these clusters represented independent introductions of the virus into Australian crocodile farms or the parallel evolution of separate viral lineages requires further investigation. The groups observed on the ML tree were also evident in NeighborNet (Figure 6B), although the Nile crocodilepox virus sequence was not included as the relationship was more distant, as indicated in the ML analysis (Figure 6A). However, a much closer evolutionary relationship was observed in gene-level phylogeny (Figure 7), where there were two distinct clusters generated among complete coding sequences of the DNA polymerase gene sourced from saltwater crocodiles and demonstrated the highest level of amino acid sequence identity (>99%).

A high degree of similarity was observed among the *crocodylidpoxvirus* genome sequences, ranging from 83.5% to 99.9%. A much higher degree of sequence similarity was observed among the SwCRV sequences (>97.0%) and, therefore, very low genetic distance was identified among SwCRV genome sequences (Appendix A). SwCRV-1, Clade-I exhibited a sequence similarity between 98.1% to 99.9% from eight lesions across three farms, whereas Clade-II, composed of four viral genome isolates from Farm 1 and one from Farm 2, which presented a very high intra-group average similarity of ≥99.0%. Clade-III included only SwCRV-2 sequences obtained from three different crocodile farms (Farms 1, 2 and 4) and not surprisingly presented lower sequences similarity (97.032%) compared to other SwCRV. The percentage similarity and distance between each sequence pair is listed in Appendix A.

### 3.4. Evidence of Inter-Farm Genetic Recombination among SwCRV

To better understand the inter-farm and/or inter-subtype likely recombination within SwCRV, recombination analyses were initially performed using seven different detection methods contained in the RDP4 program [31]. Using these methods, a large number of potential recombinations (*n* = 24) were detected within the SwCRV genomes (Appendix A). Among them, five detected recombinations (event 1, 2, 6, 9 and 16) showed robust signal/support for recombination events within SwCRV and did not show any likely errors arising from the sequencing alignment and subsequent recombination detection methods (Appendix A). Interestingly, the most substantial support for recombination was detected among SwCRV genomes isolated from four different farms (recombination event 1: RE1); where three sequences (F3a-F3c) were identified as a potential minor parent and two sequences (F4a and F1e) as a potential major parent. This recombination region overlapped from 28,110 to 51,457 and contained the genes encoding for the hypothetical protein (SwCRV1-042, -43, -45 and -46), serine/threonine protein kinase (SwCRV1-047), IEV protein (SwCRV1-048), EEV envelope lipase (SwCRV1-049) and B22R-like protein (SwCRV1-051, -052 and -055). The second recombination event (RE2) overlapped from 6956 to 25,638, which corresponded to F-box domain protein, SORF2 domain protein and hypothetical protein of SwCRV; where one Farm 2 sequence (F2c) was identified as a potential minor parent and two Farm 5 sequences (F5a and F5b) as potential major parents. Importantly, recombination event 6 (RE6) seemed to overlap within three important SwCRV genes, including RNA polymerase subunit RPO132 (SwCRV1-187), IMV A type inclusion-like protein P4c (SwCRV1-188) and IMV membrane protein (SwCRV1-189) where four sequences were identified as potential minor parent (3 from Farm 2; F2a, F2d, F2e and 1 from Farm 3; F3c) and two sequences (both from Farm 1; F1a, F1d) as a potential major parent.

To ensure the recombination events did not arise from an assembly error due to the presence of different co-infecting SwCRV variants, the three recombination events (RE1, 2 and 6) were further analysed using ML-based phylogenetics, NeighborNet trees and genetic distances. The ML trees featured all 16 SwCRV sequences, including those that were not involved in the particular recombination event, and the trees were colour coded as blue for the potential minor parent, magenta for the potential major parent and orange for the recombinant sequence (Figure 8). The generated ML trees provided further confidence for a true recombination event since the identified recombinant shifted between clades with a substantial degree of bootstrap support and grouping with the major parent in the major part of the alignment and the minor parent in the minor part of the alignment (Figure 8). The groups observed on the ML trees were also evident in NeighborNet with some degree of network branching supportive to recombination events (Appendix A). A Phi test was conducted using SplitsTree and provided statistical support for the recombination events RE1, RE2 and RE6 (*p* = 0.00, *p* = 2.223 × 10^−12^ and *p* = 2.168 × 10^−5^, respectively).

To further examine these recombination events (RE1, RE2 and RE6), Geneious software was used to display variations/SNPs for these genome comparisons (Appendix A), and pairwise distances were computed for the corresponding alignments using CLC Genomic Workbench (version 9.5.4) (Appendix A). It is important to note that these analyses were a simplified way of looking at recombination potential as only major events occurring between the analysed genomic regions were identified. For instance, there were multiple variations/SNPs, which can be seen (Appendix A), particularly between potential major parents (F1e and F4a) and potential minor parents (F3a-F3c), which were further supported by greater pairwise genetic distances (>0.16) (Appendix A). Similar scenarios were displayed in the case of RE2 and RE6 (Appendix A).

## 4. Discussion

Poxvirus infection in *C. porosus* was first reported in 1992 [32] and remained a significant economic risk due to increased production times waiting for lesions to completely heal thus they cannot be seen on the finished tanned leather product. Recent studies have confirmed the presence of typical poxvirus structures in the pathological lesions using transmission electron microscopy and the sequenced genome of saltwater crocodilepox virus [3,4]. In this study, it was demonstrated that significantly different infection dynamics and pathogenic outcomes exist between two Australian crocodile farms and these observations were further defined by genomic sequencing, whereby distinctly different clades of functional genes were formed. To achieve this, 14 complete SwCRV genomes were constructed from lesions sourced from five different Australian crocodile. Using these, in addition to two previously reported SwCRV genomes (3), we established a well-supported evolutionary relationship among poxvirus sequences under the genus *Crocodylidpoxvirus*.

This study determined that the grower phase of crocodile production presents the highest risk for the development of poxvirus lesions. Although these lesions will heal [4] and not affect the skin quality, their presence delays harvest and increases the costs of production combined with the risk of more lesions developing in the meantime. Between the two farms observed, there were definite differences in poxvirus infection dynamics. Farm 2 had significantly more lesions than Farm 1 (Figure 2 and Figure 3) and the Farm 2 crocodiles were observed to have more poxvirus lesions, in both the early active, active and expulsion stages, as they approached finishing size, conceivably deferring their movement into the higher cost/unit finishing stage of production (Figure 4). Comparatively on Farm 1, the risk of new poxvirus lesions developing significantly decreased when crocodiles were >135cm in total length (Figure 4). This information must, however, be considered in the context of farmed crocodiles, which are more densely located than animals in the wild, and it is possible that this factor may contribute to poxvirus infection dynamics, and that alternate pathogenesis may be seen in wild crocodiles. The differences in infection dynamics in these two farms, which display similar husbandry practices and share wild-harvested eggs from the same collection areas, nevertheless, led us to examine the genomic variation between SwCRV genome sequences on these and other farms in Northern Australia.

We have previously demonstrated that SwCRV on Australian crocodile farms is distinct from other chordopoxviruses, and thus its reservoir is unknown [3]. Nonetheless, SwCRV has demonstrated a close relationship with Nile crocodilepox virus isolated from a different continent with no species distribution overlap between *C. niloticus* and *C. porosus*. Genomic analysis of 16 SwCRV isolates from five farms in Australia revealed a separation of these isolates into three distinct clades, supported by both the construction of a ML tree and a NeighborNet tree. Although there was a propensity for particular farm SwCRV isolates to cluster within clades, others such as those from Farm 2 distributed across the three different clades (Figure 6), perhaps highlighting the likely mode of inter-farm viral transmission. In some cases, the distribution of clades may be reflected by animal transfer arrangements between farms. For example, Farm 5 supplies crocodiles to Farm 3 and these isolates clustered together within Clade-I, albeit at a different subclade. It is unknown if the isolates sampled from Farm 3 were from crocodiles originally sourced from Farm 5 or not. Further, Farm 3 supplies Farm 4. The isolates from Farm 4 clustered with SwCRV-2 (Clade-III), but it was again unknown if this isolate was from a Farm 4-raised crocodile or not. Farms 3 and 4 also shared wild-harvest egg collection areas similar to Farms 1 and 2. The distinct difference in the pathogenesis of poxvirus lesions (Figure 4), combined with the distribution of SwCRV isolates from these farms in alternate clades, is perhaps indicative that the dominant environmental source of the poxvirus is now mostly farm-based and acquired following hatching of the crocodiles. However, further studies are needed to clarify the initial reservoir and host range of SwCRV beyond crocodile farming. Additionally, we must also keep in mind that the farmed crocodiles are more densely located than animals in the wild and that, therefore, their infection dynamics may be altered in this setting, due to the nature of the crocodile hunting and interactions with each other in the wild that this species would still be expected to undergo.

Although we observed a high degree of sequence similarity, ranging from 97.1% to 99.9% (Appendix A), and an intact set of conversed core genes amongst all SwCRV sequences, we did observe a distinct variation in the gene encoding the IMV A type inclusion-like protein P4c (SwCRV1-188) amongst the separate isolates.

The IMVA P4c protein was fragmented due to multiple insertions/deletions in one SwCRV isolate from Farms 1, 4 and 5, and four isolates from Farm 2 (Appendix A). At the gene level, this variation was dominated in the SwCRV genomes isolated from Farm 2, being present in 4 out of 5 isolates. The poxvirus P4c protein is a structural protein present on the surface of the intracellular mature virus particle (IMV) and has been demonstrated to be necessary for directing IMV into A-type inclusions (ATI), formed by the A_type inclusion protein (Atip) [33,34]. Many orthopoxviruses embed virus particles into dense bodies, called ATIs, and it is believed that this may provide environmental protection for the virion. While many notable poxviruses, including monkeypox and variola virus, contain disrupted versions of the P4c protein (or its homolog), the lack of inclusion body formation may suggest a positive infection advantage. Interestingly, recent evidence has suggested that interruption of the cowpox P4c protein enhances the pathogenicity in the lungs of mice, as well as viral replication [35]. Given that the P4c gene is fragmented in the majority of the Farm 2 SwCRV viral isolates, a farm where we see increased presentation of poxvirus lesions, as well as enhanced pathogenicity and prolonged infection, one hypothesis might be that this gene interruption is driving this phenomenon. However, in the absence of a tissue culture system for this virus, further experimentation will be required to assess multiple early active crocodile poxvirus lesions for both their P4c sequence information and the presence of inclusion bodies in their lesions, as well as following the pathogenesis of initial infection to determine its outcome.

Viral recombination can have a major impact on the emergence of new viruses and the expansion of viral host ranges, as well as increases in virulence and pathogen diversity [36,37]. It has been well documented that recombination plays a pervasive process of generating diversity in a wide range of RNA viruses, as well as in many DNA viruses [37,38,39]. The role of recombination in the case of viruses belonging to the genus *Crocodylidpoxvirus* is still not understood due to the lack of sufficient sequence data. Interestingly, using the 16 genomes generated in this study, SwCRV genomes appear to be the subject of multiple recombination events. There was a large number (*n* = 24) of potential inter-farm and/or inter-subtype likely recombination events detected among the SwCRV genomes isolated from Australian *C. porosus*. Similarly, MCV, which is distantly evolutionarily related to members of the genus *Crocodylidpoxvirus*, has also revealed the existence of large-scale recombination events between two different MCV subtypes [40,41]. It is quite possible that the role and importance of recombination as a mechanism for SwCRV evolution may have maintained a similar pathway to MCV. The availability of more sequence data, especially from wild crocodile SwCRV lesions if found, will allow more accurate determination of these evolutionary relationships to facilitate a better understanding of the diversity observed and the variability of certain biological traits such as host range and transmissibility. These are essential factors that will influence effective management and control of this economically significant virus infection for the Australian crocodile industry.

The evolutionary origins of the *Poxviridae* family remain unknown although they are a very diverse DNA viral family that is exclusively cytoplasmic replicating and able to infect reptiles, humans, birds, mammals, insects and fish. One of the newer described members of the *Chordopoxvirinae* subfamily, within the poxvirus family, is the saltwater crocodilepox virus (SwCRV), belonging to the new genus *Crocodylidpoxvirus*. The saltwater crocodile is an ancient species, having evolved from the archosauria clade that includes the dinosaurs, and further insight into the evolution of poxvirus infection in these animals may offer valuable insights into evolution of the *Poxviridae* viral family. Additionally, the wide-spread incidence of poxvirus in farmed crocodiles may also afford the opportunity to obtain further valuable insights into natural viral selection processes in an in vivo setting.

## Figures and Tables

**Figure 1 viruses-11-01116-f001:**
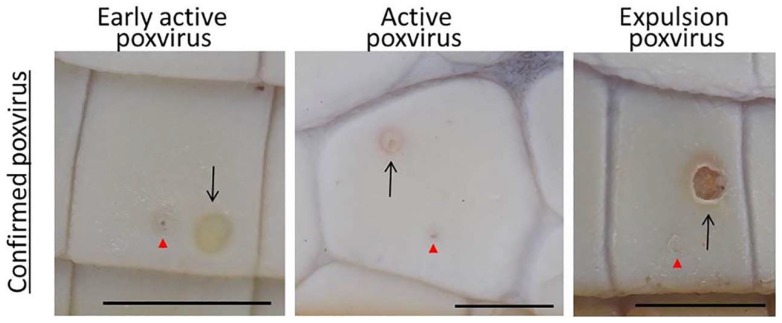
The belly skin of a juvenile saltwater crocodile showing poxvirus lesions as defined by Moore et al. [4]. Black arrows indicate the lesion tested. Red arrowheads are integumentary sensory organs (ISOs). Bar = 5 mm.

**Figure 2 viruses-11-01116-f002:**
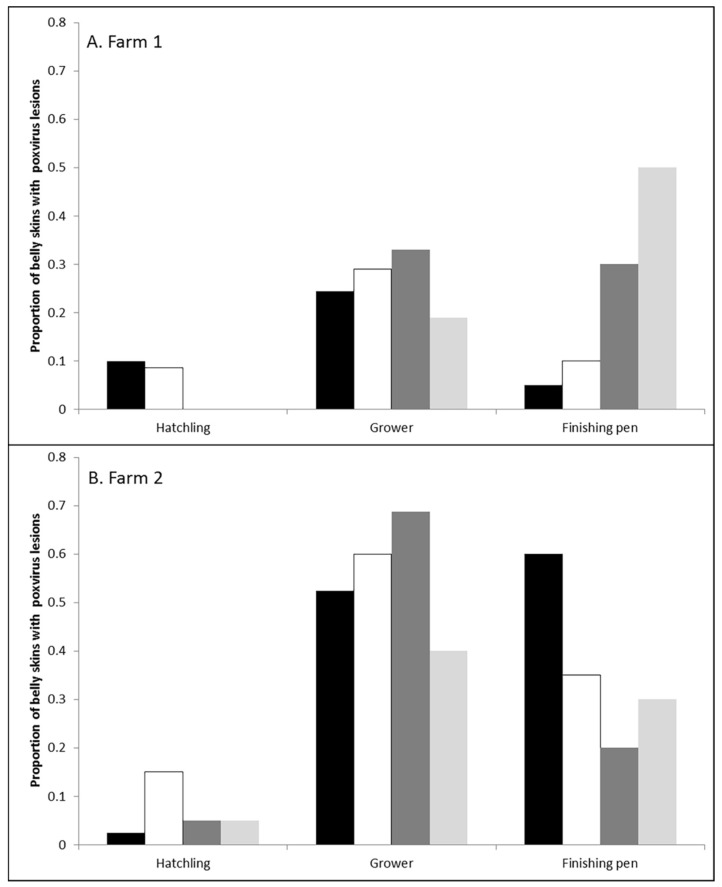
Proportion of belly skins with poxvirus lesions present on A. Farm 1 and B. Farm 2 across the three different age categories (hatchling, grower and finishing pen) and the four different sampling periods (Sampling 1 = solid black; Sampling 2 = white; Sampling 3 = dark grey; Sampling 4 = light grey).

**Figure 3 viruses-11-01116-f003:**
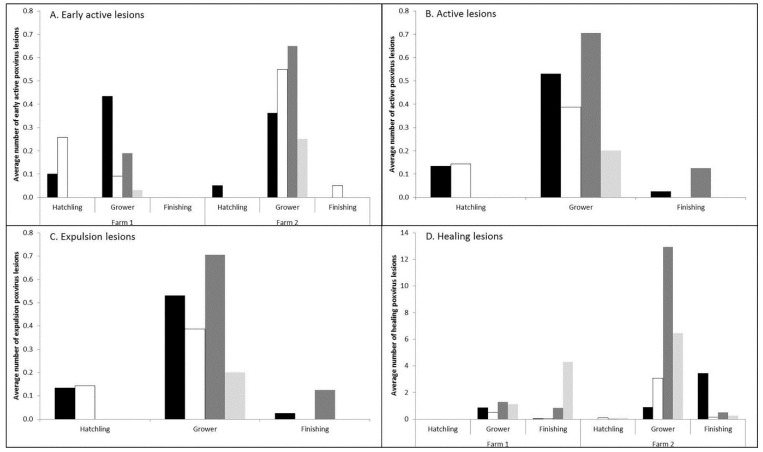
The average number of different stage poxvirus lesions in the different age categories and sampling periods (Sampling 1 = solid black; Sampling 2 = white; Sampling 3 = dark grey; Sampling 4 = light grey). Significant differences (*p* < 0.05) between farms were only observed in the number of A. early active lesions and D. healing lesions, thus only these graphs show the differences between farms.

**Figure 4 viruses-11-01116-f004:**
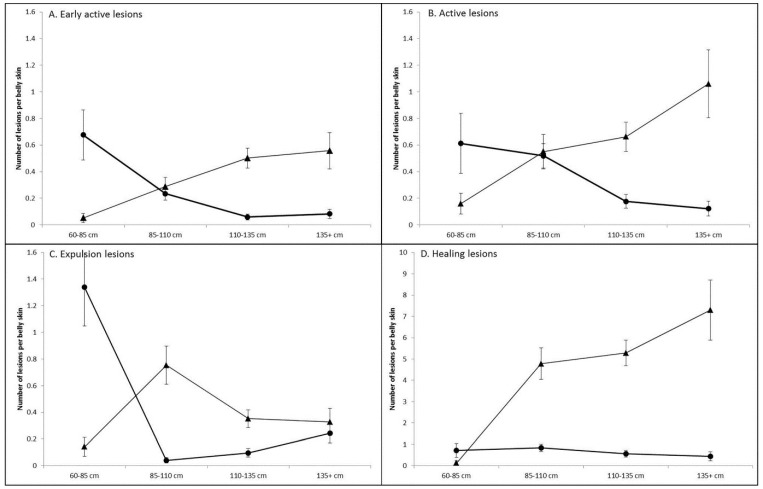
Model-adjusted means ± SE for the number of lesions per belly skin in either (**A**) Early active lesions; (**B**) Active lesions; (**C**) Expulsion lesions or (**D**) Healing lesions in the different crocodile total length size categories. Farm 1 is circles, and Farm 2 is triangles.

**Figure 5 viruses-11-01116-f005:**
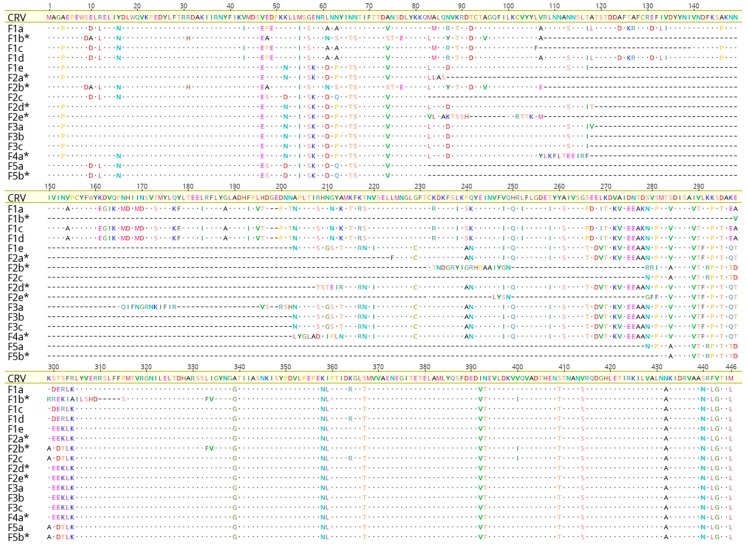
Pairwise comparison on protein sequences of IMV A type inclusion-like protein P4c under the genus *Crocodylidpoxvirus*, where IMV A type inclusion-like protein P4c of nile crocodilepox virus (CRV) was used as a reference. The fragment genes (asterisks) were aligned after concatenating the fragmented gene belonging to individual isolates. All other large gaps were related to multiple insertions/deletions.

**Figure 6 viruses-11-01116-f006:**
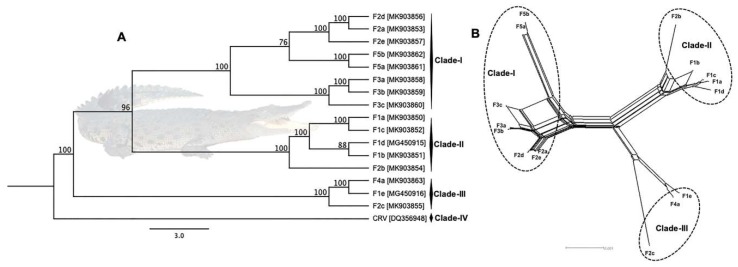
Maximum likelihood (ML) tree of crocodylidpoxviruses detected in saltwater crocodile and Nile crocodile from Australia and Zimbabwe, respectively (**A**). There was a total of 16 SwCRV genome sequences sourced from five different Australian farms (Table 4) and a Nile crocodilepox virus genome sequence sourced from Zimbabwe. The ML tree was constructed from a multiple-nucleotide alignment of ~137 kbp core region (large gaps removed) from the selected complete genome sequences of *Crocodylidpoxvirus*. The numbers on the left show bootstrap values as percentages (after 1000 replicates), and the labels at branch tips refer to original sample identification followed by GenBank accession number in parentheses. (**B**) NeighborNet tree presenting the relationship among SwCRV sequences. The network was computed using SplitsTree software. EqualAngle was employed for splits transformation. Clusters were highlighted by round dot circles.

**Figure 7 viruses-11-01116-f007:**
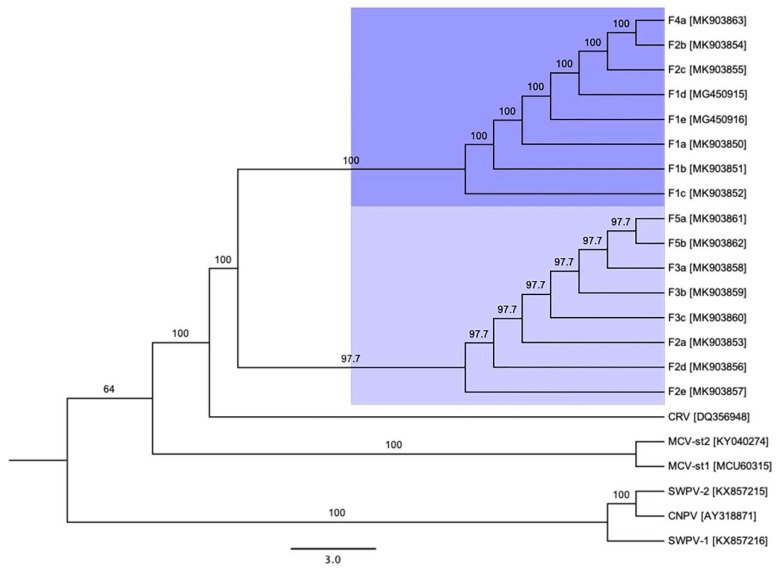
ML phylogenetic tree of the DNA polymerase gene constructed from the protein sequences of genus *Crocodylidpoxvirus*, with an addition of selected DNA polymerase gene of avipoxviruses and Molluscum contagiosum virus. The numbers on the left show bootstrap values as percentages, and SwCRV clades were highlighted using blue shading. The abbreviations for other poxviruses were used: CRV (Nile crocodilepox virus); MCV-st1 (Molluscum contagiosum virus subtype 1); MCV-st2 (Molluscum contagiosum virus subtype 2); CNPV (Canarypox virus); SWPV-1 (Shearwaterpox virus-1); SWPV-2 (Shearwaterpox virus-2).

**Figure 8 viruses-11-01116-f008:**
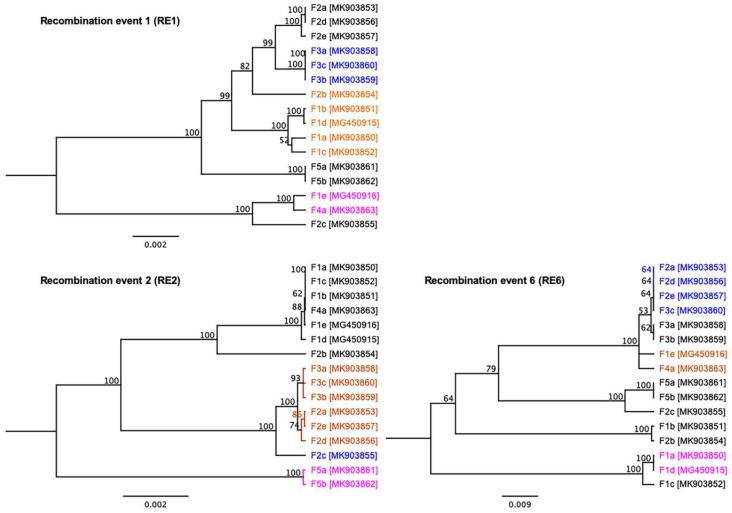
Maximum likelihood (ML) phylogenetic trees were constructed according to recombination events (RE1, RE2 and RE6). ML trees were colour coded as blue for the potential minor parent, magenta for the potential major parent and orange for the recombinant sequence. Branches were equipped with branch support values obtained from 1000 bootstrap replicates.

**Table 1 viruses-11-01116-t001:** Summary of crocodiles observed at two crocodile farms in the Northern Territory of Australia for poxvirus lesions. At each sampling, crocodiles were randomly selected from multiple pens representing 7% of hatchlings, 9% of growers and 10% of finishing pens in each randomly selected pen.

	Farm 1	Farm 2
*Sampling 1: August–October 2016*		
Hatchling	30	40
Grower	90	80
Finishing pen	20	20
*Sampling 2: January 2017*		
Hatchling	35	20
Grower	100	60
Finishing pen	20	20
*Sampling 3: August–November 2017*		
Hatchling	40	20
Grower	100	80
Finishing pen	20	20
*Sampling 4: February-–March 2018*		
Hatchling	0	20
Grower	100	40
Finishing pen	20	20
Total	575	440

**Table 2 viruses-11-01116-t002:** Predicted outcomes of extruded lesions from the belly skins of *C. porosus*. Poxvirus-like lesions were classified according to Moore et al. [4] as either early active, active or expulsion.

Sampling	Early Active	Active	Expulsion	Total Pox
*Farm 1*				
1	24	25	7	56
2	1	14	2	17
3	10	14	0	24
4	3	4	2	9
*Farm 2*				
1	10	20	8	38
2	15	13	2	30
3	6	1	0	7
4	4	2	0	6
Totals	73	93	21	187
% correct assignment	67%	84%	100%	82%

**Table 3 viruses-11-01116-t003:** Regression coefficients of poxvirus lesions stages. Correlation coefficients (*r*) are presented in parentheses on the lower diagonal. ** *p* < 0.01, *** *p* < 0.001, n.s. is non-significant (*p* > 0.05).

		Response Variate			
		Early Active	Active	Expulsion	Healing
**Explanatory variate**	Early active		1.5 ± 0.03 ***	1.37 ± 0.04 ***	n.s.
	Active	1.27 ± 0.01 *** (0.61)		1.08 ± 0.02 ***	1.13 ± 0.02 ***
	Expulsion	1.11 ± 0.01 *** (0.33)	n.s. (0.27)		n.s.
	Healing	1.02 ± 0.004 *** (0.17)	1.01 ± 0.005 ** (0.18)	n.s. (0.08)	

**Table 4 viruses-11-01116-t004:** Summary of origin, sequencing, mapping and genome statistics of 16 SwCRV isolates used in this study.

Farm ID	Sample ID	Total Reads	Total Nucleotides	Mean Read Length	Coverage	Genome Size (bp)	GC Content (%)	ITRs in the SwCRV Genome			GenBank Accession Number	Number of Annotated Genes	References
								Position in Sense-Strand	Position in Antisense-Strand	Length			
**Farm 1**	F1a	2,182,778	464,144,061	229.47	171.75	187,468	62.00	1-1140	187,468-186,329	1140	MK903850	216	This study
	F1b	625,406	171,737,498	289.27	367.73	187,223	62.00	1-1622	187,223-185,602	1622	MK903851	218	This study
	F1c	2,929,464	562,334,027	218.46	77.96	186,383	62.00	1-902	186,383-185,482	902	MK903852	213	This study
	F1d	2,263,362	580,836,186	256.63	1905.90	187,976	61..90	1-1700	187,976-186,277	1700	MG450915	218	Sarker et al. 2018
	F1e	770,348	260,017,678	267.43	476.58	184,894	62.20	1-1254	184,894-183,641	1254	MG450916	215	Sarker et al. 2018
**Farm 2**	F2a	755,168	180,551,171	260.08	268.82	187,295	62.00	1-945	187,295-186,351	945	MK903853	215	This study
	F2b	684,886	183,346,358	286.82	508.91	187,334	62.00	1-1655	187,334-185,680	1655	MK903854	217	This study
	F2c	676,468	185,563,388	287.19	663.98	184,469	62.30	1-1617	184,469-182,853	1617	MK903855	214	This study
	F2d	732,772	206,557,607	286.75	981.20	187,619	62.00	1-1291	187,619-186,329	1291	MK903856	213	This study
	F2e	2,665,954	556,343,609	222.95	101.11	185,923	62.00	1-882	185,923-185,042	882	MK903857	211	This study
**Farm 3**	F3a	1,199,052	312,997,159	272.30	1019.39	187,648	62.00	1-906	187,648-186,743	906	MK903858	215	This study
	F3b	721,906	201,313,232	287.37	938.56	187,549	62.00	1-1633	187,549-185,917	1633	MK903859	216	This study
	F3c	774,274	213,662,759	286.23	699.52	187,293	62.00	1-926	187,293-186,368	926	MK903860	215	This study
**Farm 4**	F4a	1,613,140	303,995,638	197.83	1130	185,168	62.20	1-1682	185,168-183,487	1682	MK903863	215	This study
**Farm 5**	F5a	631,050	173,473,167	286.76	560.34	186,462	62.10	1-877	186,462-185,586	877	MK903861	212	This study
	F5b	1,877,842	352,661,729	199.85	467.89	186,876	62.00	1-932	186,870-185,939	932	MK903862	213	This study

## Data Availability

All data used within this publication are available within text and supplementary files. Sequencing data used to assemble the genome are available in Table 4. The complete genome sequences of 14 SwCRV and associated datasets generated during this study were deposited in GenBank under the accession numbers MK903850 to MK903863.

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
