# Peer review of "Crocodilepox Virus Evolutionary Genomics Supports Observed Poxvirus Infection Dynamics on Saltwater Crocodile (Crocodylus porosus)"

_viruses, 2019, doi:10.3390/v11121116_

Round 1
Reviewer 1 Report
The authors study was designed to understand poxvirus infection dynamics, followed by developing a comprehensive sequence profile of a set of representative Saltwater crocodilepox virus genomes to identify likely evolutional history, genetic diversity and inter-farm genetic recombination patterns across five different crocodile farms located in Northern Australia.
Major concerns
​
• I suggest a figure with photographs of crocodiles belly skin (cited in the line 97) showing arrows in early active, active and expulsion lesions to facilitate the understanding of readers about lesions classified according to Moore, et al. ​
• In figure 4, the authors perform a Maximum likelihood tree (Fig4-A) with full genome sequence of crocodilepox viruses. However, due to large variations and gaps I suggest perform an alignment of genome sequences excluding the ITRs and use the ~100 kb highly conserved central region of genomes.​
• Authors should clarify in the discussion some points about the A-type inclusion body (ATI). A-type inclusion protein gene encodes de matrix protein of ATI, and the p4c gene that encodes a protein that directs IMVs into ATIs. Please quote this paper in your study: Okeke, M. I., Adekoya, O. A., Moens, U., Tryland, M., Traavik, T., & Nilssen, Ø. (2009). Comparative sequence analysis of A-type inclusion (ATI) and P4c proteins of orthopoxviruses that produce typical and atypical ATI phenotypes. Virus Genes, 39(2), 200–209. doi:10.1007/s11262-009-0376-8​
• As the authors emphasize the deletions in P4c gene and its possible influence on crocodylidpoxviruses, I suggest the histology of lesions with HE (could evidence the presence of ATIs in cells) and transmission electron microscopy of lesions (could show the presence of viral particles in ATIs).​
• Authors should consider the following questions in the discussion: Could the crocodile farm environment negatively select viruses in inclusion bodies? Could the high density of animals in the same place facilitate the transmission of viruses outside an environmentally resistant structure? What are the differences between the environment of crocodile farms and their natural habitat?​
Minor concerns
1- The scientific names of animals and bacterium species should be italicized. Lines (4; 18; 46; 57; 58; 197, 202, 405). The name is commonly written out in full when it first appears in the abstract and then abbreviated in the rest. The name is again written out in full when it first appears in a subsequent section of the paper (typically, the introduction) and is then abbreviated upon further use. E.g. - Lines 57; 58 rewritten: “Poxvirus lesions have been reported in a number of different crocodilians including Caiman crocodilus fuscus, Caiman crocodilus yacare, C. porosus, Crocodylus johnstoni and Crocodylus niloticus.”​ 2- The scientific names of viruses should be italicized: ​ • Line: 17 -“…genus Crocodylidpoxvirus”​ • Line 40 - “Poxviridae family”​ • Lines 44-45 – “…belongs to the genus Crocodylidpoxvirus, a member of the subfamily Chordopoxvirinae in the family Poxviridae…”​ • Line 457- “Poxviridae family”​ • Line 464- “Poxviridae viral family”​ 3- In Figure 4 subtitle, add the word genus or “Crocodylidpoxvirus “should not be italicized and not capitalized the first letter. ​ E.g. “ Maximum likelihood (ML) tree of genus Crocodylidpoxvirus detected in saltwater ….” or “ Maximum likelihood (ML) tree of crocodylidpoxviruses detected in saltwater…”​ 4- I believe the authors should rethink the following statement “…natural environment …” in line 85. Crocodile farms are not a natural environment. ​ 5- Line 269: “…with an average coverage range from 77.96 to 1905.90.” Please, clarify in the text if the values 77.96 to 1905.90 refer to average length (in nucleotides) or redundancy (77.96x to 1905.90x) .​ 6- Please, clarify in text the information’s about DNA polymerase gene. E.g : Is a member of the B family of replicative polymerases. Is correlated to the vaccinia virus E9L gene.​ 7- DNA polymerase is a highly conserved gene. Include more poxvirus species in Supplementary Fig. S3 tree. I suggest that this data be part of the main text.​
Reviewer 2 Report
The authors report the complete genome sequence of 14 saltwater crocodilepox virus (SwCRV) isolated from three Australian saltwater crocodile farms. Comparison with the two previously known SwCRV genomes shows that these viruses cluster in three clades. The finding of the three clades in the same farm suggests transmission between farms. The authors indicate small variations in the genes identified, ranging from 211 to 218. They detect 24 recombination events and some of these events cause the fragmentation of the P4c gene (a protein involved in virion embedding into A-type inclusion bodies - ATI). The discussion is focused on the P4c gene and the authors propose that variations in this gene may account for enhanced pathogenicity and increased prevalence of the virus in one of the farms.
While the report of the genomic sequences is interesting to address genetic variability of SwCRV, the analysis of the proteins predicted to be active or truncated in different isolates is not adequate, and further analysis should be provided. Specifically, the conclusion that variations in the P4c protein may cause different pathogenicity and prevalence of these viruses is not supported by the data presented. Additional evidence for the formation of ATIs and/or the inclusion of mature virions in the ATIs after infection with different isolates should be provided.
Specific comments:
A summary of the methodology used for viral DNA extraction and genome sequencing should be provided to avoid the need to find previous publications.
An analysis of the recombination events is presented (Table S1), and further detailed analysis is shown for recombination events 1, 2 and 6. The text indicates the genes contained within these regions, but it does not specify whether these recombination events cause the truncation or inactivation of these genes. This information is important since it would help the identification of genes that may influence pathogenesis, in addition to the P4c gene.
While the absence of the P4c gene has been proposed to influence viral pathogenesis in other poxviruses, the information provided or known for SwCRV is not sufficient to make these conclusions. This is the major weakness of the study, and the authors must address several relevant questions to support their conclusion:
(1) Poxviruses may form ATIs, which are mainly composed of a specific ATI protein. Is this ATI protein present or truncated in SwCRV?
(2) Are ATIs formed in cells infected with SwCRV? The electron micrographs of infected crocodile tissues published by the authors (Sarker et al. Scientific Reports 8:5623 2018) do not show ATIs. If ATIs are not formed in SwCRV infections, the discussion on whether the P4c protein is truncated in different isolates is irrelevant.
(3) The IMV ATI protein P4c is important for embedding the mature virions within the ATI. The authors indicate in the Discussion (lines 427-429) that ‘P4c protein is … necessary for the formation of ATIs’ and the Abstract (lines 31-32) that ‘P4c is … necessary for A-type inclusion formation’. This is wrong since P4c is not necessary for the formation of ATIs, but for inclusion of mature virions within ATIs.
(5) If SwCRV forms ATIs in infected cells … do they contain mature virions? The authors should determine whether the different isolates sequenced form ATIs and virions are embedded within ATIs. Otherwise, the arguments on the truncated forms of the P4c protein are not conclusive.
(6) The analysis of the amino acid sequence of the P4c protein in different isolates is not clear (Fig. S2). This Figure should be in the manuscript, not presented as a supplementary Figure. Fragmented genes are marked with an asterisk, but this is not consistent. For example, F1e shows has a fragment deleted but it is not marked like F1b*. A similar inconsistency occurs between F2c and F2b*/F2d*, and between F5a and F5b*, both showing long deletions. Most important, this analysis should be done in comparison to a conserved and active P4c protein encoded by other poxviruses. The consensus generated after comparison of all SwCRV P4c proteins may not correspond to the wild type, active protein. As presented, the Figure assumes the variation is always a ‘deletion’ that truncated the active protein. Maybe there are ‘insertions’ in some strains that inactivate the protein and these sequences are included in the consensus shown in the Figure.
Reviewer 3 Report
The manuscript entitled “Crocodilepox virus evolutionary genomics supports observed poxvirus infection dynamics on saltwater crocodile (Crocodylus porosus)” presents analyses of investigations at crocodile farms affected by Crocodilepox virus and examines relationships among isolates through genomic analyses. The topics of this manuscript are interesting and relevant to the poxvirus field. I think the presentation of the analyses and results is good, but could be improved before its acceptance/publication.
While authors indicate that husbandry practices in the two farms included in the study, the introduction of new individuals to the farms (in the different pens) could be contributing to the prevalence differences of lesions at the different stages (e.g., introducing susceptible/not previously infected animals a particular pen could explain the presence of early stages of the lesions). It would be interesting to report if individuals were introduced to pens before/between samplings. Similarly, it would be interesting to know if animals were transferred between pens as they grew and how isolated these ‘populations’ are maintained. More information regarding the husbandry practices would be relevant to the interpretation of the observations; e.g., how long are crocodiles kept in each pen, how are they moved from pen to pen, etc. Thus, the link between fragmentation of the P4c gene and the presentation of the disease is not fully supported by the present analysis and needs more investigation ideally in a closed experimental setting.
Does the categorization into four size-related groups (P8 – Figure 3) correspond to a thresholds utilized for husbandry/management of the animals? Please explain the reasoning for determining the size ranges of each category.
Other comments:
P1.L18: use italics for ‘Crocodylus porosus’
P1.L22: Italics for ‘C. porosus’
P2.L46: use italics for ‘Crocodylus porosus’
P2.L48: use italics for ‘Crocodylidpoxvirus’
P2.L57-59 (and throughout the document: Use italics for scientific names.
P2.L61: Use parenthesis around ‘4’ after ‘Moore et al.’
P2.L63: superscript for “2” in “mm2”
P2.L61-67: Indicate how long does each of the lesion stages last.
P3.L100: correct degree sign in “-20oC”
P3.Table 1: Indicate the number of individuals present in each pen at the time of sampling. Clarify if any
P5.L200: Use parenthesis around ‘15’ after ‘Isberg et al.’
P7. Figure 2: Panels B and C look exactly the same and they seem to only show one Farm, but it is unclear which one.
